# Healthy as a Horse? Characterising the UK and Ireland’s Horse Owners, Their Horses, and Owner-Reported Health and Behavioural Issues

**DOI:** 10.3390/ani15030397

**Published:** 2025-01-31

**Authors:** Wendy Leah Watson, Jill R. D. MacKay, Cathy M. Dwyer

**Affiliations:** The Royal (Dick) School of Veterinary Studies and The Roslin Institute, Easter Bush Campus, Scotland EH25 9RG, UK; jill.mackay@ed.ac.uk (J.R.D.M.); cathy.dwyer@ed.ac.uk (C.M.D.)

**Keywords:** recreational horse owners, UK and Ireland, behaviour, welfare issues

## Abstract

This study considers the owners of recreational horses in the UK and Ireland and the prevalence of welfare issues in these horses. A questionnaire was distributed to 1501 owners/managers to gather information about the owners, their horses, and reported behavioural and welfare problems. The typical respondent was a 45-year-old female (98%) owning a gelding (65%), which was a sport horse or mountain and moorland breed type. Most horses were over 5 years of age, and the most common activities were hacking (69%) and flatwork (49%). The most frequent welfare issues reported included lameness (26%), handling problems (11%), antisocial behaviours (9%), and abnormal oral behaviours (9%). Older horses were significantly more likely to experience lameness, while sport horse breeds were more prone to abnormal oral behaviours like wood chewing and crib biting. The associations between horse characteristics (such as age and breed) and welfare issues were, weak however. Therefore, other factors, such as the owner’s knowledge, attitudes, and management practices, may also be important in the incidence of these issues. This underscores the need for further research into how management practices affect horse welfare.

## 1. Introduction

The horse population in the United Kingdom and Ireland is substantial, with estimates ranging from 600,000 to 1 million animals in the UK [1] and a population of 82,010 in Ireland in 2019 [2]. Despite these large numbers, there is still a lack of understanding of the welfare risks and prevalence of welfare problems among these horses. Welfare is defined as the ability of an animal to cope with their environment, and, for managed animals, their environment is largely controlled by their caregivers [3]. Good welfare involves meeting the nutritional, environmental, health, and behavioural needs of an animal, integrated through their impact, both positive and negative, on mental state, e.g., the Five Domains [4].

It is important to understand who owns or manages horses in the UK and Ireland because owners/managers can have a significant impact on their horse’s welfare [5,6,7,8]. There are also many different types of ownership and management situations for horses, such as those who part share or loan their horses (e.g., where the horse’s care may be delivered by someone other than the owner for some or all of the time). Studies investigating horse welfare have identified a number of prominent management issues, such as poor disease prevention, lack of owner knowledge regarding the welfare needs of horses, fear and stress involved in horse riding and handling, the inability of owners to recognise pain behaviour, obesity, and inadequate feeding practices [6,7,8,9,10,11]. Therefore, there may be horses that live in suboptimal conditions and may be subject to constant anxiety, pain, and fear. To better support horse owners/managers and to improve welfare, we require a comprehensive understanding of who recreational horse owners are and the choices they make regarding management.

Horses have inherent traits, such as the need for social interaction, freedom of movement, and constant access to suitable forage [12,13,14,15,16,17], which affect their welfare. Alongside management, it is also important to consider horse characteristics, such as sex, breed, age, and types of activity routinely engaged in [18,19,20,21,22,23,24,25], to evaluate potential welfare issues. It is likely that both management and handling, along with horse characteristics, affect welfare [5,6,7,8,11,12,26,27]. Thus, it is valuable to understand the attributes of the horse, its owner, its management, and its usual use.

The purpose of this study was to answer four main questions: Who owns and/or manages horses in the UK and Ireland? What are the characteristics of horses in the UK and Ireland? What is the prevalence of health and welfare issues among horses and ponies in the UK and Ireland and are there any associations between horse characteristics such as age, height, sex, activity, breed type, and health and behavioural issues? Our null hypothesis was that there would be no associations between horse characteristics and their health and behaviour.

## 2. Materials and Methods

### 2.1. Ethical Approval

All research included in this study was approved by the University of Edinburgh’s Human Ethics Review Committee (HERC) HERC_316_19.

### 2.2. Questionnaire Design

The questions were primarily devised using peer-reviewed literature on welfare concerns associated with different horse management practices such as those questions used in the studies of Hockenhull and Creighton [5] (Q5, Q19, and Q55 in our questionnaire) and similar questions to Visser and Van Wijk-Jansen [7] can be found in Appendix A of our questionnaire. Please see these questions in the questionnaire in the Appendix A. The questionnaire was initially piloted through a focus group of horse-owning members of staff (*n* = 6) at the Royal (Dick) School of Veterinary Studies. Piloting established the validity of the questions and no questions were changed/adapted. The finalised version of the questionnaire consisted of 9 sections containing fixed-response, multiple-choice, and Likert-type questions. The questionnaire (a copy of the full questionnaire is provided in the Appendix A) covered four main categories: owner demographics, horse demographics, horse management, and horse health and behaviour. Respondents were asked to answer the management and demographic questions about one specific horse or pony, for which they made the majority of management decisions, and to subsequently answer questions for the same animal throughout the questionnaire.

There were a total of 60 questions asked on the questionnaire (see the Appendix A for a copy of the survey). An ‘Other’ category, in the form of a free text box, was included as a choice for most multiple-choice questions to capture any options not included in the pre-listed responses. Some questions allowed the respondents to select multiple answers, whereas others were limited to a single response from a list.

Health and behavioural variables were combined for analysis in order to provide a suitable sample size by adding and renaming related categories (Table 1).

There were a number of questions where the boundaries of the categories had some overlap (Table 2). These questions were exclusive-response (radio button) questions, but the label description themselves had overlap, due to human error. For example, in Question 10 ‘How many years of experiences do you have with managing horses or ponies?’, someone who had managed horses for 5 years could choose the option ‘2–5 Years’ or ‘5–8 Years’ correctly, and there was no guidance as to which option would be more appropriate. We elected to retain these questions for analysis while acknowledging this as a potential source of noise in the data. As the data are ordinal, we judged that the rank difference between groups was still meaningful, even if there was some noise around the boundaries, for example, Q.26, it is not likely that a 5-year-old horse would be misclassified as a 20-year-old horse. We considered this a more appropriate and ethical use of the data rather than removing these questions from the analysis entirely. There was no way of identifying a respondent who might fall into the double categories, so we had no method of cleaning the data post hoc.

The questionnaire was hosted on Online Surveys (Jisc, Bristol, UK, v.2) and was open from 14 January to 24 October 2019.

### 2.3. Participants and Recruitment

The questionnaire was disseminated via social media platforms such as Facebook, Twitter, and the Horse and Hound Online Forum. Internet-based questionnaires have been previously used successfully to research equine management methods [28,29]. Only one response per IP address was recorded. Exclusion criteria included non-UK or Irish postcodes, partial postcodes, ‘not known’ postcodes, and any respondent under the age of 16 years old (*n* = 30). Respondents 16 years of age or older were considered appropriate to include in the analysis because the UK Animal Welfare Act [30] stipulates that people over the age of 16 are legally responsible for an animal that they own or manage. All participation in the questionnaire was voluntary.

### 2.4. Data Cleaning and Analysis

Data were exported from Online Surveys (Jisc, Bristol, UK, v.2) to Microsoft Excel 2021 (Microsoft Corporation, Redmond, WA, USA) for cleaning. Continuous variables were tested for normality and analysed accordingly.

#### 2.4.1. Predicting Health and Welfare Issues

We were interested in exploring whether there were specific horse characteristics associated with a given animal developing a series of health and welfare issues: lameness, GI issues, hoof problems, handling issues, abnormal oral behaviours, antisocial behaviours, and weaving. For each of these response variables, answers were coded as binary responses such that 0 = no occurrence and 1 = occurrence of the issue. The horse characteristics available were horse age, sex, height, breed type (sport type versus all other breed types), and activity (any affiliated competitions, some unaffiliated competitions, and non-competitive activities) as described in Table 3 below. These variables were combined and re-grouped in (Table 3) to facilitate analysis in R. We regrouped the ‘Sports horse’ type based on an established understanding of the horse breeds that are commonly involved in higher-level equine competition or classed as sport horses [31].

The descriptive characteristics of the horse population used in the binary logistic regression analysis are shown in Table 4.

#### 2.4.2. Model Building

Each variable was fitted as a fixed effect as per Gelman’s [32] range of definitions, as they were expected to be constant across individuals and interesting in their own right. Each model was run as a binary logistic regression estimated using Ordinary Least Squares regression, where 95% Confidence Intervals (CIs) and *p*-values were computed using a Wald z-distribution approximation.

#### 2.4.3. Model Selection

For each model, we used a backward stepwise regression where the model was built first as a full additive model and each model’s cumulative explanatory power was examined via Tjur’s R^2^ [33] and the alpha level of each explanatory variable. Explanatory variables were removed from the model where the alpha level did not reach a threshold of *p* = 0.05, and the most parsimonious informative model is presented herein for each of the health and welfare issues.

These analyses were performed in R [34] and made use of ‘tidyverse’ [35], example, and ‘easystats’ [36]. Models were fitted via the ‘stats’ package and we estimated relations via the ‘model based’ package [37].

## 3. Results

### 3.1. Who Manages Horses in the UK and Ireland?

In total, 1531 participants responded to this questionnaire. After exclusion criteria were applied, there were 1501 usable responses, resulting in a completion rate of 98%. Not all respondents answered each question fully. The demographic data for the respondents are shown in Table 5 below.

The respondents were mostly female (98%) and had a median age of 45 years (range: 16–83 years). The majority resided in England and Scotland (56% and 29%, respectively).

Of the 78% of respondents who were willing to disclose their income, 74% earned over £35,000. Most of the respondents (76%) did not earn an income from equine-related activity, which suggests that this sample is largely a recreational horse-owning population.

Respondents managed a median of 2 horses, although this varied from 0 to 135. Most respondents had experience of managing horses for 14 or more years (80%), and 98% were currently managing a horse. In total, 94% of the respondents indicated that they owned the horse they reported on in the questionnaire, 1% did not own the horse they reported on, and 2% did not own the horse they reported on in the questionnaire, but had total financial responsibility for it. Another 1% did not own the horse they reported on in the questionnaire but had some financial responsibility for it, and 1% had part loan of the horse reported on in the questionnaire (responsible for the upkeep of the horse). Moreover, 61% of horses had been managed at the same location for more than 2 years. However, 10% of horses had been managed for less than 6 months in the current location. Just over half of the respondents (51%) did not insure their horse or pony for veterinarian expenses (Table 6).

### 3.2. What Are the Main Characteristics of Horses in This Sample?

The characteristics of the horses owned or managed by the respondents are shown in Table 7. Most respondents (79%) answered for a horse standing over (144 cm) tall rather than a pony (≤144 cm). Geldings (61%) were more commonly reported on than mares (38%), and few respondents reported on intact stallions (0.74%). Most horses in the questionnaire were aged between 5 and 20 years (79%), with less than 5% of horses aged under 5 years and 16% older than 20 years.

The horses reported on were from a range of different breed types (Table 7). The most commonly stated horse breed types were sport horse types (16%), mountain and moorland types (15%), cob types (15%), and thoroughbred types (14%).

Respondents reported that they had engaged in at least 12 different activities with their horses during the past month (Table 7) with both affiliated and non-affiliated activities being represented. Hacking and structured schooling on the flat were the two most popular activity types with 69% and 49% of respondents, respectively, engaging in these activities with their horses. Groundwork, such as lunging (44%), hand walking (39%), and structured schooling (jumping 25%), and competing in unaffiliated show jumping (16%) and dressage competitions, both unaffiliated (20%) and affiliated (8%), were also selected by respondents.

### 3.3. What Are the Health and Welfare Concerns of Horses and Ponies in the UK and Ireland?

When asked if their horses had any health or behavioural issues in the past six months, respondents reported that just over half of horses aged 14 plus were lame (54%) compared to 26% of horses aged between 10–14 years or 20% of horses aged 10 or younger (Table 8). Only 6% of horses aged 5 or younger were reported to have GI issues, but more than a quarter of older horses were (Table 8). Approximately a quarter of the thoroughbred horse types (25%) and sport horse types (23%) were reported as having had GI issues compared to 16% of mountain and moorland types and 7% of cobs (Table 8).

Over a third of horses aged between 10–14 years of age were reported to have displayed antisocial behaviours (37%) and abnormal oral behaviours (42%) in the past six months. However, only 6–11% of young horses (5 years or less) were reported as showing these behaviours. A quarter of thoroughbred types (24%) and sport horse types (22%) were reported as having had handling issues, compared to mountain and moorland or cob types (10% and 8%, respectively; Table 8). Antisocial behaviours were reportedly below 20% for most breeds except for sport horse types (21%). Abnormal oral behaviours were reported to be very low in mountain and moorland and cob type horses (5%), whereas the reported prevalence was 28% in thoroughbred horse types and 23% in sport horse types.

### 3.4. Health and Behavioural Concerns of Horses and Ponies in the UK and Ireland

Infectious disease was rare in the overall population, reported by less than 1% of respondents. More than a third of respondents reported that their horse had hoof and/or leg issues in the past 6 months, with 26% of horses reported as lame and 11% of owners reporting their horse as having hoof problems. GI issues were reported in 9% of the horses

Behavioural issues, related to handling, were reported by 11% of the respondents. Abnormal oral behaviours and antisocial behaviours, which included stereotypic-type behaviours, were also seen in 9% of horses in the questionnaire, but weaving was rare (1.4%).

### 3.5. Coefficients for the Models

The following results describe the coefficients for the models predicting lameness, hoof issues, GI issues, handling issues, and antisocial behaviours from horse characteristics or activity.

### 3.6. Lameness

The best model to explain lameness used age and activity as explanatory variables. This model’s explanatory power was weak (Tjur’s R^2^ = 0.05). The model intercept, corresponding to a horse <5 years old which did not participate in any competitions, estimated a probability of 13% that the horse would be lame (OR = 0.15, 95% CI [0.07, 0.30], *p* < 0.001). Horses that were 10–14 years old were more likely to be lame (OR 2.51, 95% CI [1.27, 5.59], *p* = 0.014) and horses 14+ were also more likely to be lame (OR = 4.23, 95% CI [2.17, 9.27], *p* < 0.001) than younger horses. Competing at both affiliated level (OR = 0.6, 95% CI [0.41, 0.87], *p* = 0.007) and unaffiliated level (OR = 0.53, 95% CI [0.38, 0.73], *p* < 0.001) was associated with a reduced likelihood of lameness (Table 9) below.

### 3.7. Hoof Problems

Hoof problems were best predicted in a model combining sex and activity, but again, the model’s explanatory power was weak (Tjur’s R^2^ = 0.01). The model intercept, corresponding to a gelding which did not participate in any competition, predicted a probability of 15% of hoof problems in the last 6 months (OR = 0.17, 95% CI [0.14, 0.22], *p* < 0.001). Comparatively, mares had a reduction in the probability of hoof problems (OR = −0.68, 95% CI [0.47, 0.97], *p* = 0.034). Competing at the unaffiliated level reduced the probability of hoof problems (OR = 0.48, 95% CI [0.47, 0.97], *p* = 0.002) and competing at an affiliated level was associated with a reduction in the probability of hoof problems (OR = 0.47, 95% CI [−0.26, 0.801.35], *p* = 0.008) (Table 10) below.

### 3.8. Gastrointestinal Issues

Breed type was the only significant predictor of GI issues, and the model’s explanatory power was weak (Tjur’s R^2^ = 0.01). The model intercept, corresponding to non-sport horse breed types, suggested a probability of 6% that these horses had gastrointestinal issues (OR = 0.07, 95% CI [0.05, 0.09], *p* < 0.001). Sport horse breed types were more likely to experience GI issues (OR = 2.01, 95% CI [1.40, 2.93], *p* < 0.001) than other breed types. (Table 11) below.

### 3.9. Handling Issues

Handling issues were best predicted through a model combining age and breed type. The model’s explanatory power was weak (Tjur’s R^2^ = 0.02). The intercept, corresponding to a <5-year-old horse of the non-sport horse breed type, estimated a probability of 14% that there would be handling issues (OR = 0.16, 95% CI [0.08, 0.29], *p* < 0.001). Comparatively, a 14+-year-old horse had a reduced probability of handling issues (OR = 0.38, 95% CI [0.21, 0.75], *p* = 0.003) and sport horse breed types had an increase in the probability of handling issues (OR = 2.13, 95% CI [1.53, 3.00], *p* < 0.001) (Table 12) below.

### 3.10. Abnormal Oral Behaviours

Abnormal oral behaviours were best predicted by a model incorporating age and breed type. The model’s explanatory power was weak (Tjur’s R^2^ = 0.04). The model’s intercept, corresponding to a <5-year-old horse which was a non-sport horse breed type, estimated a probability of 10%, that it showed abnormal oral behaviours (OR = 0.12, 95% CI [0.06, 0.22], *p* < 0.001). Horses that were between 5 and 10 years of age were comparatively less likely to show abnormal oral behaviours (OR = 0.41, 95% CI [0.21, 0.83], *p* = 0.011), and horses over 14 were also less likely to show abnormal oral behaviours (OR = 0.21, 95% CI [0.10, 0.43], *p* < 0.001). However, there was no statistically significant difference in the 10–14 year age category (OR = 0.62, 95% CI [0.33, 1.23], *p* = 0.152). Sport horse breed types were more likely to show abnormal oral behaviours (OR = 3.13, 95% CI [2.10, 4.73], *p* < 0.001) compared to other breed types (Table 13) below.

### 3.11. Weaving

Breed type was the only significant predictor of the horse expressing weaving behaviour. The model’s explanatory power was weak (Tjur’s R^2^ = 0.01). The model’s intercept, corresponding to a non-sport horse breed type, estimated a probability of <0.01% of weaving in the past 6 months (OR = 0.00, 95% CI [0.00, 0.01], *p* < 0.001). If the horse was a sport horse breed type, this was associated with an increase in the probability of weaving (OR = 10.3, 95% CI [2.97, 64.64], *p* = 0.002) (Table 14) below.

### 3.12. Antisocial Behaviours

Activity level was the only significant predictor of the probability of showing antisocial behaviours. The model’s explanatory power was weak (Tjur’s R^2^ = 0.004). The model’s intercept corresponded to horses that did not participate in competition and estimated an 8% probability of antisocial behaviours (OR = 0.08, 95% CI [0.07, 0.11], *p* < 0.001). Competing at the unaffiliated level was associated with an increased likelihood of showing antisocial behaviours (OR = 1.67, 95% CI [1.11, 2.49], *p* = 0.012), but there was no impact of competing at an affiliated level (OR = 1.48, 95% CI [0.90, 2.36], *p* = 0.111) (Table 15) below.

## 4. Discussion

In this paper, we report on the characteristics of the owners of recreational horses in the UK and Ireland, the types of horses that they own or manage, and the prevalence of equine health and welfare issues in this population (as reported by respondents). We also describe the underlying factors that may contribute to the health and welfare of these horses. Previous works [12,19,20,21,22,24,25,27] have suggested that there may be an interaction between horse type, activities routinely engaged in, and the incidence of welfare issues, as well as the interactions between these attributes and horse management. Therefore, we characterised the contribution of horse demographic variables and activities to commonly reported health and welfare issues.

### 4.1. UK and Irish Horse Owner Characteristics

The typical owner in our study was female, based in England, with a median age of 45 years. Our study, and others [5,38,39] have demonstrated a higher prevalence of female horse ownership, although another study found similar proportions of male and female horse owners [40]. In general, male respondents are often underrepresented in animal welfare questionnaire populations, possibly because women score higher on animal welfare attitudes than men, and a female bias exists in this type of research [41,42]. Our respondents were found to be well-educated with approximately a third of the sample population having completed an undergraduate degree and almost a quarter having completed a master’s degree. They were also relatively affluent with a third of the respondents reporting an income of between £35,000–£74,999. This is more than the median UK income, estimated at £30,378 in the year the questionnaire was administered (2019) [43]. These results are similar to the larger population in England and Wales as census results for 2019 found that 34% of the average population had more than a bachelor’s degree [44]. This suggests that our sample population were equivalent to the population average for educational attainment but more affluent than the general population. Therefore, the barriers to improving horse welfare in our study may not necessarily have been related to a financial inability to provide good welfare for the horse or the horse owner’s lack of capacity to seek equine knowledge. However, respondents reported that they had little formal equine education and therefore may have sought horse welfare knowledge from a range of sources, as reported in a previous study [7]. Lack of knowledge is a commonly proposed reason for welfare challenges, as in the study by Hemsworth et al. [6], who suggested that horse welfare issues were mostly due to horse owners’ lack of knowledge or ignorance regarding horse management.

In the current study, hacking was disclosed by the respondents as the primary activity undertaken with the horse, and the majority of respondents did not receive an income from equine-related activity. Less than 10% of the respondents in our study engaged in higher-level competition, such as affiliated dressage or showing (general). This suggests that our respondents were likely to be recreational horse owners, and this corresponds with previous findings that reported that the majority of the respondents were also leisure horse owners [39,40]. Therefore, it is possible that the reason that just over half of the horses in this study were not insured for vet expenses is that they owned recreational horses rather than competition horses. A study by Stowe et al. [45] found that horse owners who competed were more likely to insure their animals compared to those owners who did not compete. Further investigation is required to better understand the factors that contribute to these types of decisions.

### 4.2. Characteristics of the Horses in the Sample Population

The majority of equids in our study were horses as opposed to ponies, 154 cm or taller, geldings, and between 5 and 20 years of age. There was no particular preference for breed types as sport horse types, mountain and moorland types, cob types, and thoroughbred types were equally popular. These findings are similar to other scoping studies completed in the UK in which thoroughbreds/thoroughbred crosses and native/native crosses were also popular breed choices [38,39]. In another study, older horses and ponies, over the age of 15 years, were found to be in a higher proportion (29%) in the UK [40].

There was a lower proportion of female vs. male horses in our study and almost no respondents kept entire males (stallions). Riders’ preferences for geldings over mares have been previously observed, although there is limited support for the existence of sex-related differences in horse behaviour [46]. Although Aune et al. [46] did uncover some general differences in behaviour between mares and geldings, when ridden, no sex-related differences in behaviour were found, suggesting that riders may avoid mares based on perceived rather than actual behavioural reasons.

### 4.3. Health and Welfare Challenges in Horses in the UK and Ireland

Owner-reported lameness was the most common welfare issue reported in our study, which has previously been shown in other studies [19,20,22,24]. In our study, the most common age range of horses was between 10 and 14 years of age and was found to be a significant contributor to the incidence of lameness, with a higher odds ratio of lameness in older horses compared to younger horses. Age and lameness issues have previously been linked in dressage and endurance horses [19,20]. Age, height of horse, and lunging practices were all found to be associated risk factors for lameness in dressage horses [19]. However, the authors of those studies do also suggest that it was likely that management and exercise regimes could also be contributing factors in these results. Lameness prevalence has also been reported in single-breed studies, e.g., hind-end lameness appears to be associated with sire, age and height in Icelandic horses [22] and young 3-year-old American quarter horses in training were found to develop lameness, over a period of six months, due to specific hoof structure elements [24]. Other studies have also found associations between breed type and activity [24,25].

In our study, we did not see an association between breed type and lameness. However, we found that horses used in the past month, in both unaffiliated and affiliated competitions, were less likely to have reported lameness issues and hoof problems than horses that had not competed. These findings are congruent with a study by Visser et al. [12] who found that riding school horses and recreational horses were more likely to have lameness issues compared to competition horses. However, our outcome may also be an artefact of the questionnaire method used in the study, which asked owners to report on activities carried out in the past month. It is important to note that our data are derived from owner-reported conditions which may be susceptible to owner interpretation and bias. In another study, Visser et al. [12] suggested that competition owners were likely to be more knowledgeable about horses’ welfare needs, which may explain why, in their sample, competition horses were less likely to be reported to have lameness issues. Conversely, Dyson and Pollard [47] suggest that because there is such a high frequency of lameness in riding school and sport horses, many riders could be habituated to abnormal behaviours from very early on in their riding career and not readily able to recognise abnormalities. However, our study did not allow us to disentangle these potential influencing factors.

Our study found that sport horse types were twice as likely to have been reported as having GI issues (colic and gastric ulcers) in comparison with other breed types. There is a link between GI issues, such as colic, and abnormal oral behaviours, for example, crib-biting and windsucking in horses [48]. Sport horse types were almost twice as likely to exhibit abnormal oral behaviours compared with other breed types and ten times more likely to weave than other breed types. A study of young thoroughbred and part thoroughbred horses found that 35% of the population had some abnormal or re-directed behaviour [49]. Thoroughbreds and other sport horse breed types may be predisposed to these behaviours due to genetics or specific management practices, such as inappropriate early weaning practices, lack of social contact and confinement, lack of provision of foraging opportunities, and a diet high in concentrated feed [49,50]. It is possible that the breed impacts observed in our study could be confounded by the management strategies used and the use of different breeds.

In our study, horses aged 10–14 years old were less likely to have been reported to show abnormal oral behaviours compared to younger horses. Young horses typically show more oral behaviours than adults, including oral play, manipulation of substrates, and oral appeasement gestures (mouth clapping), which appear to be part of the normal developmental patterns of horses [51,52]. It has been suggested that the origins of stereotypic behaviour usually manifest within one month of a foal being weaned [53], when both the social and nutritional environments are altered. Although there may be a genetic link to stereotypic behaviour in horses, as suggested by our data showing an association with breed type, it is possible that both environment and genetics play a key role in the manifestation of these behaviours [54,55].

Just over 10% of the horses in our study were reported to have behavioural problems that made them more challenging to handle. Also, we found that young horses and sport horse types were significantly more likely to have handling issues than older or other breed types. Antisocial behaviours, such as aggression towards people or other horses, were linked with participating in unaffiliated competitions in our study but not with breed type. These types of behavioural issues may also be an indicator of pain that owners may not readily recognise and therefore overlook in day-to-day care, particularly with regard to saddle fit [11,47,56], tacking up, mounting technique, and lameness [51]. The authors of another study [57] found a higher percentage of horses with handling issues (63%) compared to our study. In the aforementioned study [57], the authors asked about the frequency of handling issues, whereas we asked for a specific time frame for handling issues observed in the past 6 months, which could account for a lower percentage of handling issues being observed in our study.

In this study, we found some links between the characteristics of the horse and their welfare, but the statistical model’s explanatory power was weak. Horse health and welfare issues are likely to be complex and related to many interrelated factors, such that a simple combination of horse characteristics is unlikely to explain all of the variations in each issue. Instead, a more detailed understanding of owner knowledge, attitudes, perceptions, horse management, and human behaviour barriers will be more helpful in tackling welfare issues in horses in the UK and Ireland.

## 5. Conclusions

Our study provides information on the types of owners and horses that encompass the recreational horse-owning population in the UK and Ireland.

We report a relatively low rate of health and welfare issues in the sampled population, which may reflect the method of data collection, which relies on owner identification and recollection of issues, although the main issues reported are similar to those seen in other studies. Whilst horse age, breed type, and type of activity routinely engaged in were seen as contributing factors to the health and behavioural issues found in our study, they had only a weak explanatory power over the issues reported. It is also likely that other aspects, such as the owner’s attitudes, knowledge, and management strategies, may affect the incidence of these welfare issues.

The inability to effectively model various equine health conditions reflects the study design. The type of questionnaire we distributed was primarily useful for gathering descriptive information. More targeted studies, such as a study of owner-reported health issues, would provide clearer criteria on which to base such determinations.

Thus, further investigation to uncover the causal factors leading to reductions in horse welfare is needed, if progress is to be made to improve the welfare of recreational horses. This study provides information on the prevalence of owner-reported health and behaviour conditions, which could help provide a focus for future work. In a follow-up study, we examine the impact of management practices and strategies, with the goal of investigating how these different attributes may influence horse welfare.

### Limitations

Horse owners who were more concentrated on the welfare of their horses may have been more likely to have completed our questionnaire than those who may be less interested in the welfare of their animals, which has been a criticism of similar studies [7], and this may have been exacerbated by the length of the survey. Participation in the questionnaire was voluntary and self-reported, and therefore, it could be considered that our study, along with similar studies, may not provide a clear representation of the general horse-owning population [7]. This was an online questionnaire and was thus inaccessible by default to those without technological ability or online access. It was estimated in 2023 that over 7 million households in the UK had no Internet access [58], and while we do not know what size this population is in relation to the equine-owning population, this does represent a bias in online questionnaire research [59].

There was a high proportion of Scottish respondents (29%) in our results, possibly because the questionnaire was disseminated from the Royal (Dick) School of Veterinary Studies, which is based just outside of Edinburgh. Our sample was educated to the same standard as the wider UK population [44]; however, it could be considered that the reason that there were many affluent horse owners in our sample could be because horses are expensive animals to buy and keep, and therefore it may be that horse owners are more likely to be reasonably well off. Another possible limitation of the study could be that all the health and behavioural issues are reported from the horse owner’s perspective and may not reflect a professional evaluation by a veterinarian.

Combining equine health conditions, which could have different risk factors, to create composite binary outcomes, may also explain the weak relationships uncovered in the study and similar limitations could be associated with the approach taken to combine breed types and activities for the explanatory variables. The risk of misclassification of horse age, and other categories, was outlined in the methodology and we acknowledged this as a potential source of noise in the data. However, because the data are ordinal, we felt that the rank difference between groups was still important to report, in addition to the ethical use of the data, even if there was some dataset noise around some of the question category boundaries. We believed that the respondents were capable of accurately determining what they considered to be the correct answer to these questions.

## Figures and Tables

**Table 1 animals-15-00397-t001:** Definitions of new groups of variables and the constituent categories (see the full questionnaire) that were reallocated to the new groups to facilitate analysis.

**Health Issue**	**Re-Grouped As**
Strangles/equine flu	Infectious diseases
Lameness, laminitis (acute and chronic), bowed tendon, and arthritis	Lameness
Abscess and thrush	Hoof problems
Colic and gastric ulcers	Gastrointestinal issues (GI issues)
**Behavioural Issues**	
Crib biting, chews or tears rugs (in stall), drinks water excessively, eats bedding, wind sucking, repetitively licks objects, i.e., stall wall, and wood chewing in stall or on fence	Abnormal oral behaviours
Difficult with the farrier or trimmer, difficult to lead or turnout, ‘pulls’ faces or fidgets when being tacked up, tries to bite or kick when being groomed, and tries to bite or kick when being tacked up	Handling issues
Pins back ears or lunges out towards people at feeding time, ‘pulls’ faces when people approach or walk by the stable, repeatedly kicks the stall wall/door, shows aggression to other horses, shows aggression to people, and turns away when people enter the stall	Antisocial behaviours
Weaving	Weaving

**Table 2 animals-15-00397-t002:** Questions with numerical categories which are not mutually exclusive.

Question	Category
Question 10 ‘How many years of experience do you have with managing horses or ponies?’	Less than 2 years 2–5 years 5–8 years 8–14 years 14+ years Prefer not to say
Question 22, ‘How long have you managed this horse (regardless of whether or not you own it?)’	2 years or less 2–5 years 5–10 years 10+ years N/A Other
Question 23, ‘How long has your horse or pony been in its current location?’	Less than 6 months 6 months–2 years 2–5 years 5–10 years 10–16 years 16+ years
Question 26, ‘What is the approximate age of this horse?’	2 years or less 2–5 years 5–10 years 10–14 years 14–20 years 20+ years
Question 40, ‘During the past week what was the size of the area your horse was (most commonly) turned out on?’ (NB: This question was not a part of the analysis in this paper)	Less than an acre 1–2 acres 2–5 acres 5 or more acres Do not know

**Table 3 animals-15-00397-t003:** Definitions of new groups of explanatory variables and the constituent categories (see the full questionnaire) that were reallocated to the new groups to facilitate logistic regression analysis in R.

Explanatory Variable	Re-Grouped As
**Activity**	
Hacking, hand walking, lunging, structured schooling (flatwork and jumping), showing (general and in hand), round pen work, use of a treadmill and hotwalker, mounted games, breeding, TREC, hunting, vaulting, and ponying (leading another horse whilst mounted).	All Other Activities
Dressage, driving, eventing, endurance, and show jumping (affiliated)	Affiliated Activities
Dressage, driving, eventing, endurance, and show jumping (unaffiliated)	Unaffiliated Activities
**Age**	
2 years or less	<5 Years
2–5 years	<5 years
5–10 years	5–10 years
10–14 years	10–14 years
14–20 years	14 plus years
20 plus years	14 plus years
**Breed Type**	
Sport horse type, thoroughbred type, warmblood type, Irish Draught type, or hunter type	Sport horse types
Arabian type, cob type, mountain and moorland type, crossbreed, draft (heavy horse) type, miniature horse type, Western type, or other	All other types

**Table 4 animals-15-00397-t004:** Descriptive characteristics of horse population used in binary logistic regression.

Characteristic	Type	Frequency	Percentage
Activity	All Other Activities	932	62.09
	Unaffiliated Competition	342	22.78
	Affiliated Competition	227	15.12
Breed Type	Other	771	51.37
	Sport type	730	48.63
Age	<5 Years	77	5.14
	5–10 Years	415	27.70
	10–14 Years	428	28.57
	14 Plus Years	578	38.58
Sex	Gelding	917	61.30
	Intact stallion	11	0.74
	Mare	568	37.97

**Table 5 animals-15-00397-t005:** Overview of key demographics of 1501 horse managers in the UK and Ireland who responded to the questionnaire. Values are total numbers with % of respondents in parentheses (*n* = 1501 unless otherwise stated).

**Gender**	**Count/Percentage**
Female	1475 (98%)
Male	24 (2%)
Prefer not to say	2 (<1%)
**Age**	
Median	45 years
Interquartile Range	21 years
Min Age	16 years
Max Age	83 years
**Country of Residence**	
England	836 (56%)
Scotland	433 (29%)
Republic of Ireland	125 (8%)
Wales	59 (4%)
Northern Ireland	45 (3%)
Isle of Man	3 (<1%)
**Highest Educational Qualification**	
No Education	20 (1%)
High School or Equivalent	328 (22%)
Higher National Certificate/Diploma	255 (17%)
Undergraduate	478 (32%)
Masters	336 (22%)
PhD	59 (4%)
Other †	25 (2%)
**Household Income (Pre-Tax)**	
Less than £20,000	141 (9%)
£20,000 to £34,999	253 (17%)
£35,000 to £49,999	241 (16%)
£50,000 to £74,999	262 (17%)
£75,000 to £99,999	139 (9%)
£100,000 or more	141 (9%)
Prefer not to say	324 (22%)
**Income from Equine-Related Activity**	
Yes, entirely	97 (6%)
Yes, partially	262 (17%)
Not at all	1123 (76%)
Prefer not to say	19 (1%)

† Other includes artist, paralegal, master saddler, etc.

**Table 6 animals-15-00397-t006:** Overview of key management characteristics for 1501 horse managers in the UK and Ireland who responded to the questionnaire. Values are total numbers with % of respondents in parentheses (*n* = 1501 unless otherwise stated).

**How many horses/ponies do you currently manage?**	**Count/Percentage**
Median	2 horses
Interquartile Range	2 horses
(Min)	0
Max	135
**Years of experience managing horses or ponies**	**Count/Percentage**
Less than 2 years	11 (1%)
2–5 years	50 (3%)
5–8 years	65 (4%)
8–14 years	174 (12%)
14 or more years	1197 (80%)
	**Total = 1497**
**Do you currently manage (make the majority of decisions for) a horse or pony?**	
Yes	1464 (98%)
No	32 (2%)
Do Not Know	5 (0.33%)
**Do you own the horse or pony (reported on in the questionnaire)?**	
Yes, I own	1410 (94%)
No, I do not own	18 (1%)
Do not own but financially responsible	37 (2%)
Do not own but partially responsible financially	12 (1%)
Part loan (responsible for all upkeep)	17 (1%)
†† Other	4 (0.27%) **Total = 1498**
**How long have you managed this horse/pony?**	
2 years or less	291 (19%)
2–5 years	431 (29%)
5–10 years	373 (25%)
10 plus years	402 (27%)
††† Other	4 (0.28%)
**How long has the horse/pony been kept at the current location?**	
Less than 6 months	155 (10%)
6 months-2 years	422 (28%)
2–5 years	453 (30%)
5–10 years	285 (19%)
10–16 years	125 (8%)
16 plus years	9 (4%)
	**Total = 1449**
**Do you insure your horse/pony for veterinary expenses?**	
Yes	719 (48%)
No	766 (51%)
Prefer not to say	16 (1%)

†† Other—not known; ††† Other—‘Only recently started part loaning the horse, but known him for 3–4 years from a riding school I worked at’.

**Table 7 animals-15-00397-t007:** Overview of key demographics for horses owned by respondents in the UK and the Republic of Ireland including horse vs. pony, sex, age range, breed type, and activity. Values are total numbers with % of respondents in parentheses (*n* = 1501 unless otherwise stated).

**Type of Equid**	**Count/Percentage**
Horse	1177 (79%)
Pony (144 cm or under)	315 (21%)
¶ Other	5 (0.33%)
	**Total = 1497**
**Sex**	
Gelding	917 (65%)
Mare	568 (38%)
Intact Stallion	11 (0.74%) **Total = 1496**
**Age Range**	
2 years or less	10 (1%)
2–5 years	67 (4%)
5–10 years	415 (28%)
10–14 years	428 (29%)
14–20 years	342 (23%)
20 Plus Years	236 (16%)
	**Total = 1498**
**Height (Cm)**	
124 cm and under	32 (2%)
123–144 cm	294 (20%)
145–153 cm	254 (17%)
154–163 cm	403 (27%)
164–173 cm	421 (28%)
174 cm plus	93 (5%)
	**Total = 1497**
**Horse Type**	
Sport horse type	246 (16%)
Mountain and Moorland type	228 (15%)
Cob type	219 (15%)
Thoroughbred type	214 (14%)
Warmblood type	177 (12%)
Crossbreed type	136 (9%)
Other type	73 (5%)
Arab type	53 (4%)
Irish Draught	52 (3%)
Hunter type	41 (3%)
Draft horse	32 (2%)
Western type	26 (2%)
Miniature horse	4 (0.27%)
**Proportion of the Activity Choices Owners Participated in with Their Horse (In Past Month)**	
Hacking	1039 (69%**)**
Schooling (Flatwork)	739 (49%)
Lunging	659 (44%)
Hand walking	578 (39%**)**
Structured Schooling (Jumping)	379 (25%)
Dressage (Non-affiliated)	307 (20%)
Show Jumping (Non-affiliated)	46 (16%)
Dressage (Affiliated)	116 (8%)
Showing (General)	106 (7%)
Round Pen Work	99 (7%)
¶¶ Other	155 (10%)
	**Total Responses (Multiple) = 4117**

¶ Icelandic horse under 14 hh. ¶¶ Liberty work, swimming, and long reining.

**Table 8 animals-15-00397-t008:** Counts and percentages of owner-reported horse health and behavioural issues in the past six months.

Horse Characteristic		Health Issues n (%)			Behavioural Issues n (%)	
**Age of Horse**	**Lameness**	**Hoof Problems**	**GI Issues**	**Handling Issues**	**Antisocial Behaviours**	**Abnormal Oral Behaviours**
<2–5 Years	9 (2%)	7 (4%)	8 (6%)	15 (9%)	8 (6%)	14 (11%)
5–10 Years	70 (18%)	36 (23%)	49 (37%)	55 (33%)	44 (32%)	36 (28%)
10–14 Years	101 (26%)	57 (36%)	42 (31%)	50 (30%)	52 (37%)	54 (42%)
14 Plus Years	211 (54%)	59 (37%)	34 (25%)	49 (29%)	35 (25%)	26 (20%)
**Sex of Horse**						
Male	252 (64%)	110 (69%)	82 (62%)	92 (54%)	76 (55%)	81 (63%)
Female	139 (36%)	40 (31%)	51 (38%)	77 (46%)	62 (45%)	49 (38%)
**Five of the Top Most Reported Horse Breed Types**						
Sport horse type	54 (14%)	18 (11%)	31 (23%)	37 (22%)	29 (21%)	30 (23%)
Mountain and Moorland type	56 (14%)	22 (14%)	16 (12%)	17 (10%)	15 (11%)	7 (5%)
Cob type	52 (13%)	23 (14%)	10 (7%)	13 (8%)	18 (13%)	7 (5%)
Thoroughbred type	63 (16%)	33 (21%)	33 (25%)	41 (24%)	22 (16%)	37 (28%)

**Table 9 animals-15-00397-t009:** Estimates of the coefficients for a model predicting lameness from age and activity; the reference category (intercept) was a horse less than 5 years old which did not participate in any competitions.

Parameter	Coefficient	Standard Error	95% CI Lower	95% CI Upper	Odds Ratio (Exponential of Coefficient)	Odds Ratio 95% CI Lower	Odds Ratio 95% CI Upper	z	*p*
(Intercept)	−1.86	0.36	−2.63	−1.22	0.15	0.07	0.30	−5.22	<0.0001
Age 5–10 Years	0.55	0.38	−0.16	1.35	1.73	0.86	3.87	1.43	0.152
Age 10–14 Years	0.92	0.37	0.24	1.72	2.52	1.27	5.59	2.47	0.014
Age 14 Plus Years	1.44	0.37	0.77	2.23	4.23	2.17	9.27	3.93	<0.001
Activity Unaffiliated Competition	−0.64	0.16	−0.97	−0.32	0.53	0.38	0.73	−3.86	<0.001
Activity Affiliated Competition	−0.50	0.19	−0.88	−0.14	0.60	0.41	0.87	−2.68	0.007

**Table 10 animals-15-00397-t010:** Estimates of the coefficients for a model predicting hoof issues from sex and activity; the reference category (intercept) corresponds to a gelding which did not participate in competitions.

Parameter	Coefficient	Standard Error	95% CI Lower	95% CI Upper	Odds Ratio (Exponential of Coefficient)	Odds Ratio 95% CI Lower	Odds Ratio 95% CI Upper	z	*p*
(Intercept)	−1.75	0.11	−1.98	−1.54	0.17	0.14	0.22	−15.38	<0.001
Sex—Intact stallion	−13.52	434.07	NA	18.62	0.00	NA	122,343,873.34	−0.03	0.975
Sex—Mare	−0.38	0.18	−0.75	−0.03	0.68	0.47	0.97	−2.12	0.034
Activity Unaffiliated Competition	−0.74	0.24	−1.22	−0.29	0.48	0.29	0.75	−3.10	0.002
Activity Affiliated Competition	−0.75	0.28	−1.35	−0.22	0.47	0.26	0.80	−2.64	0.008

NA = not applicable.

**Table 11 animals-15-00397-t011:** Estimates of the coefficients for a model predicting gastrointestinal issues from breed type; the reference category (intercept) category corresponds to non-sport horse breed types.

Parameter	Coefficient	Standard Error	95% CI Lower	95% CI Upper	Odds Ratio (Exponential of Coefficient)	Odds Ratio 95% CI Lower	Odds Ratio 95% CI Upper	z	*p*
(Intercept)	−2.71	0.15	−3.02	−2.43	0.07	0.05	0.09	−18.20	<0.001
Sport horse breed type	0.70	0.19	0.33	1.07	2.01	1.40	2.93	3.71	<0.001

**Table 12 animals-15-00397-t012:** Estimates of the coefficients for a model predicting handling issues from age and breed type; the reference category (intercept) corresponds to non-sport horse breed type, less than 5 years old.

Parameter	Coefficient	Standard Error	95% CI Lower	95% CI Upper	Odds Ratio (Exponential of Coefficient)	Odds Ratio 95% CI Lower	Odds Ratio 95% CI Upper	z	*p*
(Intercept)	−1.83	0.31	−2.47	−1.25	0.16	0.08	0.29	−5.93	<0.0001
Age 5–10 Years	−0.48	0.33	−1.10	0.19	0.62	0.33	1.20	−1.48	0.140
Age 10–14 Years	−0.64	0.33	−1.26	0.03	0.53	0.28	1.03	−1.94	0.052
Age 14 Plus Years	−0.96	0.33	−1.58	−0.29	0.38	0.21	0.75	−2.93	0.003
Sport horse breed type	0.76	0.17	0.43	1.10	2.13	1.53	3.00	4.43	<0.001

**Table 13 animals-15-00397-t013:** Estimates of the coefficients for a model predicting oral behaviours from age and breed type; the reference category (intercept) corresponds to the non-sport horse breed type, less than 5 years old.

Parameter	Coefficient	Standard Error	95% CI Lower	95% CI Upper	Odds Ratio (Exponential of Coefficient)	Odds Ratio 95% CI Lower	Odds Ratio 95% CI Upper	z	*p*
(Intercept)	−2.15	0.33	−2.84	−1.53	0.12	0.06	0.22	−6.49	<0.001
Agee 5–10 Years	−0.90	0.35	−1.57	−0.18	0.41	0.21	0.83	−2.56	0.011
Age 10–14 Years	−0.48	0.34	−1.12	0.21	0.62	0.33	1.23	−1.43	0.152
Age 14 Plus Years	−1.57	0.36	−2.27	−0.83	0.21	0.10	0.43	−4.31	<0.001
Sport horse breed type	1.14	0.21	0.74	1.55	3.12	2.10	4.73	5.50	<0.001

**Table 14 animals-15-00397-t014:** Estimates of the coefficients for a model predicting weaving from breed type; the reference category (intercept) corresponds to a non-sport horse breed type.

Parameter	Coefficient	Standard Error	95% CI Lower	95% CI Upper	Odds Ratio (Exponential of Coefficient)	Odds Ratio 95% CI Lower	Odds Ratio 95% CI Upper	z	*p*
(Intercept)	−5.95	0.71	−7.75	−4.82	0.00	0.00	0.01	−8.41	<0.001
Sport horse breed type	2.33	0.75	1.09	4.17	10.27	2.97	64.64	3.13	0.002

**Table 15 animals-15-00397-t015:** Estimates of the coefficients for a model predicting antisocial behaviours from activity; the reference category (intercept) corresponds to horses not participating in competitions.

Parameter	Coefficient	Standard Error	95% CI Lower	95% CI Upper	Odds Ratio (Exponential of Coefficient)	Odds Ratio 95% CI Lower	Odds Ratio 95% CI Upper	z	*p*
(Intercept)	−2.48	0.12	−2.73	−2.25	0.08	0.07	0.11	−20.22	<0.001
Activity Unaffiliated Competition	0.51	0.21	0.10	0.91	1.67	1.11	2.49	2.50	0.012
Activity Affiliated Competition	0.39	0.24	−0.11	0.86	1.48	0.90	2.36	1.60	0.11

## Data Availability

Data supporting the reported results can be found at https://figshare.com/s/2421919f8ca9b0df348e accessed on 19 December 2024.

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
