# Peer review of "Healthy as a Horse? Characterising the UK and Ireland’s Horse Owners, Their Horses, and Owner-Reported Health and Behavioural Issues"

_animals, 2025, doi:10.3390/ani15030397_

Round 1

Reviewer 1 Report

Comments and Suggestions for Authors

Dear Authors,

thank you for a well-written paper. You have been very successful in getting so many answers from horse owners in the UK and Ireland. It seems that this topic raises interest and that horse owners are willing to take part in such surveys. 

I only have one question on the section 2.2 questionnaire design on page 3: In the questionnaire (supplement) it is said: "It will take approximately 10-15 mintues to complete." How do you know it actually took 20-50 min?

Author Response

Comments 1: Dear Authors,

Thank you for a well-written paper. You have been very successful in getting so many answers from horse owners in the UK and Ireland. It seems that this topic raises interest and that horse owners are willing to take part in such surveys. 

I only have one question on the section 2.2 questionnaire design on page 3: In the questionnaire (supplement) it is said: "It will take approximately 10-15 minutes to complete." How do you know it actually took 20-50 min?

Response 1: Many thanks your positive comments and for raising this question.  We have removed the comment from the Materials and Methods, that the questionnaire took 20-50 mins, as this was an estimate based on feedback from the pilot participants and we did not measure this in the main study.

Reviewer 2 Report

Comments and Suggestions for Authors

Dear Authors

Please find my feedback in the attached file. 

Author Response

Author's Reply to the Review Report (Reviewer 2)

Alterations to the text (other than deletions) have been indicated by highlighted text.

Comments 1: Introduction - Paragraph 2 "Veterinary science" is a field of knowledge and cannot require or otherwise act. Reword

Response 1: Thank you for pointing this out.  Now amended as “We require a comprehensive understanding of who recreational horse owners are, and the choices they make regarding management”. (Line 65)

Comments 2: Introduction - Paragraph 3

First sentence. Provide references; there are many.

Response 2: Thank you for pointing this out. Several references haves been added now and reference list amended accordingly (Line 68).

Comment 3: Third sentence. Provide references.

Response 3: Thank you for pointing this out. Several references added now and reference list amended accordingly (Line 72).

Comment 4: Fourth sentence. "Predicting" infers bias and that the authors have predetermined the study's outcome. Reword as a hypothetical statement. Then clearly state the aim of the study.

Response 4: Thank you for pointing this out. We have changed this to a hypothetical statement as suggested. (Line 90). We have also added the aims, as you suggested, which were outlined in the ‘Materials and Methods’.  

Comment 5: Materials and methods: 2.1 These are research aims and not part of the methods and materials. Please see the previous comment.

Response 5: Thank you for this comment and we agree.  They have been moved to the Introduction section as mentioned above (Line 80).

Comment 6: In what way were studies of Hockenhull and Creighton and Visser and Van Wijk-Jansen used? Copied, modified? Clarify

Response 6: Thank you for pointing this out. We used similar question types both from Hockenhull and Creighton (Q5, Q19, Q55) and Visser and Van Wijk-Jansen (Page 10: Section 9: Information Sources).

Comment 7: How many people were used to pilot this study?

Response 7: Thank you for this comment and apologies we forgot to include this number. There were 6 people used to pilot the questionnaire and this information has now been added to the methodology (Line 86)

Comment 8: In the questionnaire, the authors claim that this study does not identify participants. However, it does ask respondents to name their horses. This increases the possibility of identifying some participants. Furthermore, the authors describe using the IP address as part of their data validation and postcode collection in section 2.3, which again increases the risk of identifying a respondent. How was this managed to avoid the risk of identifying owners? This study does not appear to have gone through a human ethics approval or registration process, which could have identified these risks. This is interesting, as the authors mention ethics in discussion. Clarify.

Response 8: Thank you for pointing this out. The study was approved by the University of Edinburgh Human Ethics Review Committee, but this statement was inadvertently removed during the formatting of the paper. We have now reinstated this in the Materials and Methods. We were not seeking to identify participants by their horse’s names, the first part of their postcode or IP addresses. As there were 1,501 respondents we feel that this avoided any unintentional loss of anonymity as it is unlikely that respondents could be easily identified by their horse’s name or postcode. The JiSC online survey software uses IP address to prevent respondents from providing multiple responses, as reported in the paper, but this information is not provided directly to the research team and therefore this was not collected by the project team.

Comment 9: Flaws in the study the design regarding the overlap of categories for some questions, as mentioned by the authors, should be addressed as a limitation of the study in the discussion.

Response 9: Thank you for pointing this out. This limitation was discussed in section 5.1 however we now have more clearly highlighted this limitation in this section (Line 542).

Comment 10: 2.4.3. It's unclear if the authors used stepwise regression and whether this was forward or backward. Clarify.

Response 10: Done and added to this section.

Comment 11: The authors have not provided information on sample size or power calculations. It is also unclear whether the authors tested continuous variables for normality.

Response 11: Thank you for pointing this out.  We did not use power calculations because this was primarily a descriptive scoping study to understand horse ownership.  Continuous variables were tested for normality and analysed accordingly. (Line 139).

Comment 12: Results 3.1. The authors mentioned that exclusion criteria were applied. The exclusion and inclusion criteria for surveys are not described in the methods and materials. Rectify.

Response 12: Thank you for pointing this out. We have clarified in this in the Material and Methods (Line 131).

Comment 13: Table 5. Was the participant age distribution normally distributed? If so, report the mean and standard deviation. If not, report the median and interquartile range. Reporting median and mean suggests that this variable was not tested for normality.

Response 13: Thank you for this comment. The participant age was not normally distributed and this information has been adjusted to median and IQ range in Table 5. 

Comment 14: Table 6. Was the equine age distribution normally distributed? If so, report the mean and standard deviation. If not, report the median and interquartile range. Reporting median and mean suggests that this variable was not tested for normality.

Response: Thank you for this comment. We collected data for equine age in categories. Horse age range is reported in Table 7 and collected as categorical data. However, we have tested the normality of the continuous variable of how many horses are managed and modified Table 6 to include only median and IQR and min and max, so this is clear going forward.

Comment 15: 3.3 If they were "more likely" (i.e., there were statistically significant differences), then provide the P values, odds ratios and confidence intervals.

Response: Thank you for pointing this out. This section has been re-worded to reflect the descriptive nature of the data presented and to provide evidence of the prevalence of these conditions (Line 238).

Comment 16: 3.5 - 3.12. The description of the final models in this way makes it difficult for the reader to see the actual coefficient estimates and values for intercepts. I strongly suggest you put the models into a table so that the reference groups are easily understood. It is recommended that a supplementary table be included for each model that encompasses both included and rejected variables, their values, confidence intervals and p values.

Response 16: Tables have been added as requested Section 3.5 -3.12.

Comment 17: Discussion: The discussion should be markedly reduced and focused on the results and the citation of relevant comparative data, and not a broader review of the literature.

Response 17: Thank you for your comments. The discussion has been reduced following your recommendations (as outlined below) and we have focused on our results and explanations from comparative data as suggested.

Comment 18: A purpose is stated that is not in total agreement with the study questions outlined (incorrectly) in the materials and methods section. The authors have asserted that they have reported on the "typical owner." This is an unfounded assumption that asserts that the respondents to a survey are typical of the population of owners. Furthermore, the authors have not provided information on sample size or power calculations that provide evidence for this assumption.

Response 18: Thank you for your observations. Please see our response to comment 31.

Comment 19:4.1. First paragraph. Omit: "In a study of 126 Sport horses 6% were found to have days lost for show jumping competition due to non-acute (55%) or acute orthopedic injuries [22%]. Thoroughbred horses used for racing were found to be lame at different intervals during the racing season (53%), and in some cases (20%), the lameness was severe enough to impede any further racing [34]". Irrelevant; the authors' survey did not investigate the impacts of lameness on performance in UK horses.

Response 19: Agree and done.

Comment 20: Omit: "and we would hope that lame horses would not actively be used in competition". The discussion should focus on discussing the results of the study and not the hopes and aspirations of the authors.

Response 20: Agree and done.

Comment 21: Omit or abbreviate to compare your findings with what is directly relevant to your results: "Visser et al. [12] proposed that competition owners were likely to be more knowledgeable about horse's welfare needs, which may explain why, in their sample, competition horses were less likely to be reported to have lameness issues. Conversely, Dyson and Pollard [34] suggest that because there is such a high frequency of lameness in the riding school and Sports horses, many riders could be habituated to abnormal behaviors from very early on in their riding career and not readily able to recognize abnormalities.

Response 21: We have made this section shorter and clearer about how this information relates to our findings.

Comment 22: “However, as mentioned previously, there are many other possible mitigating factors, apart from age and type of activity, which can affect the incidence and prevalence of lameness and hoof problems; such as hoof conformation [30, 32], age of horse when being first being trained to saddle [32,34], height of the horse (lunging and the effect of torque) [26], workload (in show jumping horses) [33], saddle fit and rider position [37], environment (surface of ground), and arena footing [26,33]”. Omit. Extensive discussion of a negative finding (in this case) is unwarranted.

Response 22: Removed.

Comment 23: "This multifactorial causation of lameness is supported by the weak fit of our model in terms of explaining the incidence of lameness in our sample." It is not. No inference should be drawn from any of your models due to their extremely low explanatory contributions. I suggest you restrict your discussion to univariate associations like the one you report for weaving in sports horses.

Response 23: We agree and we have removed this sentence.

Comment 24: Omit: "It has been observed that riding style and management factors were associated with the prevalence of behavioural issues in mixed breed horses [24] and in a recent Hungarian study, a higher percentage of sport and race-horses displayed abnormal repetitive behaviours compared to other breeds [25). Dressage and eventing horses, which spent more time confined to their stables than horses in other disciplines, such as endurance, have also been shown to be more likely to display an in-crease in abnormal behaviours, such as wood-chewing, weaving, crib biting/windsucking and box-walking [41). Thus, although the explanatory power of our models was weak, our finding that Sport horse types were more likely to show these types of behavioural responses and to have a higher incidence of GI issues, aligns with previous studies."

Response 24: Removed.

Comment 25:  Omit: "It has been observed that riding style and management factors were associated with the prevalence of behavioural issues in mixed breed horses [24] and in a recent Hungarian study, a higher percentage of sport and race-horses displayed abnormal repetitive behaviours compared to other breeds [25). Dressage and eventing horses, which spent more time confined to their stables than horses in other disciplines, such as endurance, have also been shown to be more likely to display an in-crease in abnormal behaviours, such as wood-chewing, weaving, crib biting/wind-sucking and box-walking [41). Thus, although the explanatory power of our models was weak, our finding that Sport horse types were more likely to show these types of behavioural responses and to have a higher incidence of GI issues, aligns with previous studies."

Response 25: All removed.

Comment 26: No inference should be drawn from any of your models due to their extremely low explanatory contributions. I suggest you restrict your discussion to univariate associations.

Response 26: Thank you for your comment. We added the statement that we could not disentangle potential influencing factors in our study.

Comment 27: Omit: "This theory is supported by other studies, which connect stereotypical behaviour in adult horses with low forage intakes and restricted access to social contact between horses [44,45,46]. Although there may be a genetic link to stereotypic behaviour in horses, as suggested by our data showing an as-sociation with breed type, it is likely that both environment and genetics play a key role in manifestation of these behaviours [46,47]. Wood chewing in horses, which was also included in our description of abnormal oral behaviours, although destructive in a domestic environment, is also a natural behaviour in horses, although it may also be a precursor to crib-biting [48). The horse owner's attempts to curb these types of oral behaviors often result in further exacerbation of the behavior and poorer welfare for the horse [49)." This is a review of stereotypical behavior and nota not a discussion of your results.

Response 27: Removed.

Comment 28: Perfect: "Just over 10% of the horses in our study were reported to have behavioural problems that made them more challenging to handle. Also, in our study, young horses and Sport horse types were significantly more likely to have handling issues than older or other breed types. Antisocial behaviours, such as aggression towards people or other horses, were linked with participating in unaffiliated competitions in our study, but not with breed type. These types of behavioural issues may also be an indicator of pain that owners may not readily recognise and therefore overlook in day-to-day care, particularly with regard to saddle fit [11, 35, 49], tacking up, mounting technique and lameness [51). Other work [52], has found a higher percentage of the horses with handling issues (63%) com-pared to our study. In that study [52], the authors asked about the frequency of handling issues, whereas we asked for a specific time frame of handling issues observed in the past 6 months, which could account for a lower percentage of handling issues being observed in our study." This is an example of a much more results-focused discussion.

Response 28: Thank you for your positive comments.

Comment 29: 4.2 Markedly abbreviate. The survey executed by the authors did not evaluate the knowledge of the respondents, only their qualifications. Again, focus your discussion on your results and not conduct an extensive literature review of this subject.

Response 29: Abbreviated as requested.

Comment 30: 5.1. State the limitations of this study concisely. Do not justify or otherwise discuss the gender bias in depth.

Response 30: This was abbreviated.

Response 31: Omit: "Therefore, it could be considered that we are merely looking at the 'tip of the iceberg' with this data, compared with the overall population of the UK and Ireland". Do not draw conclusions based on limitations of your study.

Response 31: Sentence removed. For the type of data we have collected power calculations are not representative. However, we sampled an estimated 0.4% of the UK horse owning population (BETA 2019). Other rules suggest that to measure frequency we should aim for a sample of between 1,000 and 2,000 people which is the sampling rate we achieved. We therefore believe that sample size is not a limitation of this study. The sample size achieved in this study did attract positive comments by the other reviewers.

Comment 32: The rationalization of the misclassifications projects bias. Simply acknowledge that this was a flaw in the study design.

Response 32: This has been done.

Comment 33: An additional limitation is that the survey was extremely long which would have affected participation and completion.

Response 33: We agree and have added further comments to this effect since it is likely that only very motivated horse owners may have responded in full, and this may have led to a more positive bias.

Comments 34: An additional limitation is that all health and behavioral issues are reported from the caregiver's perspective and may not reflect a professional evaluation. Indeed, some interesting information published by others in the United Kingdom has shown that there's a poor correlation between owner’s perceptions of health issues and that of veterinarians following examination.

Response 34: Thank you for this comment. Sentence added in the limitations to cover this (Line 591)

Comments 35: The failure to effectively model different equine health conditions is a reflection of the study design. The type of survey used is most helpful for obtaining descriptive information. More focused studies are required to model specific health or welfare issues. For example, a study of owner-reported health issues would provide clear criteria on which to base such determinations.

Response 35: Thank you. We have added this to the conclusion. (Line 561).

Reviewer 3 Report

Comments and Suggestions for Authors

Dear authors,

your work is a very important contribution to understand correlations between welfare issues and surrounding circumstances as well as possible risk factors.

The proposed follow up with a focus on management and welfare is even more interesting.

One criticism point from my side: The questions have different time lines. For example past 3 month, last week…This must be carefully reconsidered to conceptualize the new survey. Because horses management most often presents with seasonal changes and so on.

Author Response

Comments 1: Dear Authors,

Your work is a very important contribution to understand correlations between welfare issues and surrounding circumstances as well as possible risk factors.

The proposed follow up with a focus on management and welfare is even more interesting.

One criticism point from my side: The questions have different time lines. For example, past 3 months, last week…This must be carefully reconsidered to conceptualize the new survey. Because horse’s management most often presents with seasonal changes and so on.

Response 1: Many thanks for your positive comments and helpful advice. For the review of the next version of the manuscript we have taken into account any potential issues relating to seasonal considerations.

Round 2

Reviewer 2 Report

Comments and Suggestions for Authors

Dear Authors

The manuscript is markedly improved in the methods and discussion sections - thank you The following information is requested to clarify the understanding of the models.  

Table 9. What were the reference categories for the age and activity variables?

Table 10. What were the reference categories for sex and activity variables?

Table 11. What was the reference category for breed - sports type?

Table 12. What were the reference categories for sex and breed - sports type?

Table 13. What were the reference categories for sex and breed - sports type?

Table 14. What was the reference category for breed - sports type?

Table 15. What was the reference category for the activity variable? 

Author Response

Comments: Dear Authors

The manuscript is markedly improved in the methods and discussion sections - thank you. The following information is requested to clarify the understanding of the models.  

Response: Thank you for the positive comments.  I have addressed your questions by highlighting what the reference category (intercept) corresponds to in the Table titles and outlined below. I hope this adds clarity in being able to understand the model.

Table 9. What were the reference categories for the age and activity variables?

Reference category (intercept) was a horse less than 5 years old which participated in no competition, compared to the age categories, age 5-10 years, 10-14 years, 14 plus years and unaffiliated and affiliated competitions.

Table 10. What were the reference categories for sex and activity variables?

The reference category (intercept) corresponds to a gelding which participated in no competition compared to stallions and mares and which competed in unaffiliated and affiliated competitions.  

Table 11. What was the reference category for breed - sports type?

The reference category (intercept) corresponds to non-Sport horse breed types compared to Sport horse breed types.

Table 12. What were the reference categories for sex and breed - sports type?

The reference category (intercept) corresponds to non-Sport horse breed type, less than 5 years old compared to a Sport horse breed type, age 5-10 years, 10-14 years, 14 plus years.

Table 13. What were the reference categories for sex and breed - sports type?

The reference category (intercept) corresponds to non-Sport horse breed type, less than 5 years old compared to a Sport horse breed type, age 5-10 years, 10-14 years, 14 plus years.

Table 14. What was the reference category for breed - sports type?

The reference category (intercept) corresponds to a non-Sport horse breed type compared to a Sport horse breed type.

Table 15. What was the reference category for the activity variable? 

The reference category (intercept) corresponds to horses which did not participate in competition compared to horses competing in unaffiliated and affiliated competitions.
